# Actin Bundles in The Pollen Tube

**DOI:** 10.3390/ijms19123710

**Published:** 2018-11-22

**Authors:** Shujuan Zhang, Chunbo Wang, Min Xie, Jinyu Liu, Zhe Kong, Hui Su

**Affiliations:** Key Laboratory of Resource Biology and Biotechnology in Western China, Ministry of Education College of Life Science, Northwest University, Xi’an 710069, China; 201720857@stumail.nwu.edu.cn (S.Z.); wangcb97@163.com (C.W.), xieminhezjk@163.com (M.X.), liujinyu@stumail.nwu.edu.cn (J.L.); kongzhe@stumail.nwu.edu.cn (Z.K.)

**Keywords:** actin bundles, actin-binding proteins, pollen tube, Arabidopsis

## Abstract

The angiosperm pollen tube delivers two sperm cells into the embryo sac through a unique growth strategy, named tip growth, to accomplish fertilization. A great deal of experiments have demonstrated that actin bundles play a pivotal role in pollen tube tip growth. There are two distinct actin bundle populations in pollen tubes: the long, rather thick actin bundles in the shank and the short, highly dynamic bundles near the apex. With the development of imaging techniques over the last decade, great breakthroughs have been made in understanding the function of actin bundles in pollen tubes, especially short subapical actin bundles. Here, we tried to draw an overall picture of the architecture, functions and underlying regulation mechanism of actin bundles in plant pollen tubes.

## 1. Introduction

In the sexual reproduction of angiosperms, firstly, the pollen grain hydrates and produces the pollen tube. Then, it passes through the column and reaches the micropyle via apical growth. After entering the embryo sac, the pollen tube releases two sperms. Finally, the double fertilization process is completed [1]. In this process, the polar growth of pollen tubes is a key link in the sexual reproduction of higher plants. It is of great significance for the reproduction of plants and the evolution of species.

Early studies have shown that the actin skeleton plays a vitally important role in the tip growth of pollen tubes [2,3]. The growth of pollen tubes depends on the targeted vesicle transport [4], through which vesicles containing cell-membrane and cell-wall resources are transported to the apical region. The vesicles, as well as other organelles, moving back and forth along the main axis of the pollen tube, are controlled by the myosin-actin filaments system [5,6,7,8,9]. In 1974, Condeelis was the first to discover a large number of actin filaments in pollen tubes [10]. Since then, researchers have tried various strategies to explore the microfilament skeleton in pollen tubes. It is now clear that the pollen tube is characterized by at least three different arrangements of actin filaments, which roughly correspond to the apex, the sub-apex, and the shank [11]. In the apical region, microfilaments in the form of short segments are highly dynamic [12,13], while in the subapical region, dense longitudinal actin bundles with a diverse organization emerge, such as the collar [14,15], mesh [16,17], funnel [3,18], and fringe [19]. Furthermore, the microfilament structure of this region is highly sensitive and unstable [20]. In the shank, the microfilaments consist of long, rather thick bundles, which are parallel to the growth direction of the pollen tube and extend to the base of the subapical region [21]. The material transport in pollen tubes includes both long-distance transport and short-distance transport. Long-distance transport relies on the microfilament bundles in the pollen tube shank [22,23], while short-distance transport depends on the short and highly dynamic actin bundles in the subapical region [24,25,26]. A large amount of evidence has shown that the proper organization of these actin bundles is crucial for pollen tube growth, hence, it is essential to reveal the underlying mechanisms of assembly and regulation of actin bundles. This article aims to provide a comprehensive survey covering the distribution, organization, and regulation of actin bundles in pollen tubes.

## 2. Spatial Distribution of F-actin Bundles in Pollen Tubes

The actin in vivo can be present as either a free monomer called the G-actin (globular) or as part of a linear polymer microfilament called the F-actin (filamentous) [27]. The G-actin has an asymmetric structure. Therefore, the F-actin, which is primarily composed of G-actin polymers, is considered to have an intrinsic polarity. The faster-growing end of the F-actin is called the plus end (barbed end), which tends more to bind ATP-G-actin, while the slower-growing end is called the minus end (pointed end), with ADP-G-actin dissociating slowly from the microfilament [28]. A specific phenomenon, named “treadmilling”, can be observed under certain conditions. During “treadmilling”, subunit addition and loss are dynamically balanced at the two ends of a filament with actin filaments assembling at the barbed end and disassembling simultaneously at the pointed end [29]. In cells, the F-actin is directional and highly dynamic, and its assembly is controlled by a variety of actin-binding proteins [30].

In plant cells, the F-actin has many distinct architectures regulated by a variety of actin-binding proteins, such as foci, rings, and meshwork [31,32,33,34]. One type of architecture concerns the parallel actin bundles, which have a profound influence on cell shape, division, motility, and signaling [35,36]. Noticeably, actin bundles are present in virtually all plant cells. There are masses of bundled filaments present in the cortex of epidermal pavement cells of rosette leaves, mature root epidermal cells, and trichomes [31,32], and almost all plant cell types have a mesh of subcortical actin bundles radiated from the perinuclear actin basket [31].

The pollen tube is characterized by at least two types of actin bundles, which roughly correspond to the sub-apex and the shank. In the shank, the long and thick actin bundles are parallel with the axis of pollen tubes, with some of the bundles even extending close to the subapical area but without penetrating into the subapical region [19,37,38]. Compared to the bundles in the sub-apex region, these longitudinal actin bundles in the shank are shown to be less sensitive to actin-depolymerizing drugs of lower concentration [3,14], suggesting that longitudinal actin cables are relatively stable. Consistently, in *pLat52::Lifeact-GFP* transgenic pollen tubes, when tracing the dynamics of individual actin filaments, the maximal filament length substantially increased and the severing frequency substantially decreased in comparison to that in the apical region [23]. By replacing the strong Lat52 promoter with the moderately strong Actin 3 promoter, the Baluška group discovered that the thicker F-actin bundles in pollen tube shanks are stable, but thinner F-actin bundles are dynamic, showing undulating movements [39]. Furthermore, the bundles of actin filaments in the shank exhibit uniform polarity with those close to the cell cortex, having their barbed ends oriented towards the tip of the pollen tube, while those in the cell center have their barbed ends oriented toward the base of the tube [40]. Myosin, the actin-based motor tracking with the bundles, moves exclusively towards the plus end of the filament, allowing for the transport of organelles or vesicles from the base to the tip along the cell cortex [41,42].

The actin organization in the apical domain of lily pollen was clearly revealed in 2005 by the Hepler group. They confirmed that actin bundles do exist in the sub-apex, and can form a dense cortical fringe or collar starting about 1–5 μm behind the extreme apex and extending basally for an additional 5–10 μm [19]. Subsequently, the actin fringe was observed in the sub-apex of pollen tubes among various species including *Arabidopsis*, tobacco, and lily [26,33,43,44]. Electron microscopy studies also showed shorter actin filaments in the sub-apex of pollen tubes, organized in closely packed and longitudinally oriented bundles, with some of them forming curved bundles adjacent to the cell membrane [39]. By using the advanced spinning disk confocal microscopy, it was recently clarified that the sub-apex actin structure consists of longitudinally aligned actin bundles at the cortex, which corresponds to the previously described actin fringe [19], and the inner fine actin filaments in the core of the cytoplasm [26]. These short actin bundles in the subapical region are regarded as a kind of turning point in the tube cell, because most of the larger organelles stop and reverse their movement in proximity of the actin fringe, while secretary vesicles can usually cross the actin fringe and reach the apex [11]. Recent research has shown that the short actin bundles at the cortex of the sub-apex region exhibit uniform polarity with their barbed-end anchoring on the apical membrane [7,45], and, supposedly, they serve as tracks for the barbed end directed myosin Ⅺs [46], while the internal fine actin filaments in the sub-apex are involved in regulating the backward movement of vesicles, presumably by acting as a physical barrier [26].

In summary, the two types of actin bundles, which roughly correspond to the shank and the sub-apex in pollen tubes, ensure orderly vesicle transport from the base to the tip, thus guaranteeing rapid tip growth.

## 3. Molecular Mechanisms Underlying the Formation and Regulation of Actin Bundles in Pollen Tubes

Proper actin cytoskeleton organization is important for the polarity growth of pollen tubes. Within the tubes, the assembly and disassembly of the actin filaments are promoted by many actin-binding proteins (ABPs), including nucleating, depolymerizing, severing, capping, F-actin stabilizing, and G-actin sequestering proteins [6,9,47]. Among the various ABPs, actin-bundling proteins trigger the formation of bundles consisting of several parallel actin filaments tightly packed together, and play essential roles in the pollen tube elongation [35,48]. This article will focus on six kinds of ABPs in plants, including villins, formins, fimbrins, LIM-domain containing proteins (LIMs), Crolins, and actin depolymerizing factors (ADFs). A detailed description of their biochemical properties, cellular localization, and the potential regulatory mechanisms in pollen tubes is given.

### 3.1. Villins

Plant villins, which contain multiple 15 kDa gelsolin/severin domains, belong to the villin/gelsolin/fragmin superfamily. In addition to the six gelsolin domains, villin has a head domain at the C-terminal, providing an additional microfilament binding site [49]. This superfamily can manipulate microfilaments in multiple ways, including capping, severing, promoting nucleation, and bundling, some of which are influenced by Ca^2+^.

In mice, the deletion of villins prevents Ca^2+^-induced actin fragmentation and disrupts the brush border, which means that villins are essential for mouse actin rearrangement after stimulation [50]. Mutations of villins in Drosophila induce female sterility and lead to defects of actin bundles in vegetative cells [51,52].

P-135-ABP and P-115-ABP isolated from lily (*Lilium longiflorum*) by biochemical fractionation [53,54] were identified to be the homologues of animal villins [55,56]. The two proteins organize actin filaments into bundles with uniform polarity [56,57]. And in both cases, the bundling activity was hindered by Ca^2+^/ Ca^2+^-calmodulin (CaM) [56,58]. To explore the role of P-135-ABP and P-115-ABP, researchers microinjected the corresponding antisera into root hairs. After injection, they found obvious changes, such as the disappearance of transvacuolar strands and the alteration of cytoplasmic streaming [56,59]. Furthermore, arranging the bundles in the shank region, lily villin also modulates actin dynamics in the apical region through its capping and severing activities, if there is a relatively high concentration of Ca^2+^ [60].

There are five villin-like genes in *Arabidopsis*, named *AtVLN1* to *AtVLN5* [61]. Two of them, AtVLN2 and AtVLN5, are abundant in pollen [62,63]. For AtVLN5, biochemical studies showed that it retains a whole suite of activities, including filaments bundling, barbed-end capping, and calcium-dependent severing. The absence of VLN5 does not affect the organization or amount of filamentous actin in pollen tubes because VLN2 acts in a redundant manner with VLN5 to regulate actin dynamics in the pollen tube [25,62]. The down-regulation of both VLN2 and VLN5 led to a remarkable reduction in the amount of actin filaments in the sub-apex, and the actin cables become thinner and more disorganized in the shanks. Consistently, the rate of pollen tube growth decreased in *vln2 vln5* [25]. In addition, AtVLN4 is abundant in root hairs and root hairs also perform typical tip growth. AtVLN4 can bundle microfilaments at a lower Ca^2+^ concentration, while it severs and caps microfilaments under a higher concentration of Ca^2+^. In the atvln4 mutant, the root hairs are shorter and the actin bundles in the hair cells are fewer in comparison to the wild type [64].

### 3.2. Formins

Microfilament polymerization is mainly initiated by nucleation factors. There are two classes of nucleation factors, formin and the Arp2/3 complex. Formin launches linearly arranged microfilaments, while the Arp2/3 complex forms reticular microfilaments. However, neither the growth of pollen tubes nor the development of root hairs is dramatically affected in four of arp mutants grown under normal conditions [65]. Formins, composed of a conserved, proline-rich formin-homology 1 (FH1) domain and a FH2 domain, are major actin filament nucleation factors in the pollen tube. The FH2 domain is required for actin filament nucleation, while the FH1 domain recruits profilin–actin complexes to the assembly machine [66]. Plant formins have capping, severing, and bundling functions in addition to the core nucleation activity [67]. Moreover, formins have the ability to faithfully track growing barbed ends to provide a means for the continuous elongation of actin [68].

Unlike other actin bundlers, formins have a large number of homologous variants in the model plant *Arabidopsis*. It can be divided into two groups, according to sequence similarity and conservatism. Group I has a unique N-terminal, which includes a proline-rich, potentially glycosylated extracellular domain and a transmembrane domain [69]. There are 21 formins in *Arabidopsis*, including 11 members (AtFH 1-11) in group I and 10 members (AtFH12-21) in group II [70]. Rice has 11 Class I Formins (OsFH1, OsFH2, OsFH4, OsFH8-11, OsFH13-16) and 5 Class II Formins (OsFH3, OsFH5, OsFH 6, OsFH7, OsFH12) [71].

Up to now, several plant formins, including AtFH1, AtFH8, AtFH14, AtFH16, OsFH5, and LlFH1, have been verified to bundle actin filaments [20,64,72,73,74]. Furthermore, AtFH1, AtFH3, AtFH5, and LlFH1 have roles in pollen tubes. AtFH1 is the first plant formin that regulates actin organization in pollen tubes. The overexpression of AtFH1 has been shown to result in excessive actin cables inside the tube and to induce membrane curvature at the pollen tube tip, suggesting that AtFH1 is important for tip-focused cell-membrane expansion in pollen tubes [69]. AtFH3 is another actin nucleation factor responsible for longitudinal actin cables in pollen tubes. Biochemical analysis revealed that the FH1FH2 domain of AtFH3 interacts with the barbed end of actin filaments and has actin nucleation activity in the presence of G-actin or G-actin profilin [75]. Specific down-regulation of AtFH3 lessens actin polymers in pollen grains and eliminates actin cables in pollen tubes. The disruption of longitude actin cables alters the reverse fountain streaming pattern in tube cells and, thus, the morphology of AtFH3-RNAi pollen tube appears abnormal [75]. AtFH5 is an apical membrane-anchored nucleation factor that initiates major actin filament assembly from the apical membrane. The actin fringe could not form properly and the pollen tube formed drastic twists and turns with the deletion of this protein, suggesting that AtFH5 plays important roles in the construction of actin structures in the apical and subapical regions [7]. In support of this notion, biochemical data revealed that AtFH5 is capable of nucleating actin assembly from actin monomers or actin monomers bound to profiling [76]. Recently, LlFH1, the pollen-specific formin in lily, has also been studied. It concentrates at the plasma membrane and vesicles in the apical region of pollen tubes. The overexpression of LlFH1 induces excessive actin cables in the tube tip region, and the down-regulation of LlFH1 eliminates the actin fringe [24]. Biochemical assays showed that LlFH1 FH1FH2 first nucleate actin polymerization, but then capped actin filaments at the barbed end and inhibited elongation [24]. However, in the presence of profilins, FH1FH2 of LlFH1 accelerates barbed-end elongation rather than inhibiting [24]. Collectively, it is proposed that LlFH1 and profilin coordinate the interaction between the actin fringe and exocytic vesicle trafficking during pollen tube growth [24].

### 3.3. Fimbrins

Fimbrin, also named plastin, is a well-characterized actin-bundling protein conserved in eukaryotes. It has been demonstrated that fimbrin is involved in polar growth across different species. For example, the fission yeast *Schizosaccharomyces pombe* assembles actin bundles oriented parallel to the long axis of cells for polarization. When deprived of the fimbrin homologous protein FIM1, the cells have mild polarity defects with their actin bundles appearing to be qualitatively more randomly oriented [77]. In *Aspergillus nidulans*, the disruption of the fimbrin homolog protein FIMA results in delayed polarity establishment, abnormal hyphal growth, and endocytic defects in apolar cells [78]. In mammals, it has been shown that the overexpression of plastin3 can repair neuronal defects associated with spinal muscular atrophy (SMA) disorders, such as axon length and outgrowth defects, illustrating that plastin3 plays a role in neurite outgrowth during axonal differentiation [79].

Five fimbrin-like genes, FIM1–FIM5, are present in the *Arabidopsis* genome. Microarray data predict that AtFIM3, AtFIM4, and AtFIM5 are expressed in pollens, with AtFIM4 and AtFIM5 being most abundant [80]. Only AtFIM4 and AtFIM5 have been intensively studied. AtFIM5 decorates actin filaments throughout the pollen tube, especially in the tip region [74]. When knocking out AtFIM5, researchers observed that the actin fringe was impaired in the apical region and that the longitudinal actin bundles in the shank were disorganized [81,82]. AtFIM4 is expressed only after pollen grains start hydrating. Moreover, its expression level is increased gradually with the extension of the pollen tube, indicating that AtFIM4 may play an important role in the polarity growth of pollen tubes. However, the mutation of AtFIM4 does not cause obvious phenotypes. In fim4/fim5 double mutants, the length of the pod and the number of seeds are both remarkably decreased, as a result of the defects of pollen germination and pollen tube growth. Furthermore, the degree of filamentous bundles inside the tubes reduces obviously, suggesting that AtFIM4 backs AtFIM5 to organize and maintain normal actin structures [83].

### 3.4. LIMs

Plant LIM proteins are a family of actin-bundling proteins with LIM domains interacting directly with actin filaments [84,85,86,87]. Most plant LIM proteins belong to the cystein-rich protein (CRP) subfamily [88,89,90] and have two conserved LIM domains and a long inter LIM spacer (40 to 50 amino acids) [91].

By using confocal microscopy, Thomas et al. were the first to show that *Nicotiana tabacum* WLIM1 is an actin-binding protein in plants [84]. Later, biochemical experiments revealed that NtWLIM2 directly bound to actin filaments and crosslinked the latter into thick actin bundles. The function of a lily *(Lilium longiflorum*) pollen-enriched LIM protein, LlLIM1, was also explored. Cytological and biochemical assays verified that LlLIM1 promoted filamentous actin bundle assembly and protected F-actin against latrunculin B-mediated depolymerization. Furthermore, its actin-binding affinity is simultaneously regulated by both pH and Ca^2+^ [86]. Pollen tubes with overexpressed LlLIM1 showed retarded pollen germination and tube growth as well as abnormal morphology, such as swollen tubes and multiple tubes protruding from one pollen grain. These are concurrent with the formation of an asterisk-shaped F-actin aggregation and abnormal endo-membrane structures in the apical of pollen tubes [86]. Therefore, LlLIM1 was considered to perform important roles in integrating endomembrane trafficking and growth in the apical region of pollen tubes where pH and calcium oscillate regularly.

The *Arabidopsis* genome contains six genes encoding LIM proteins, three of which are predominantly expressed in pollen: PLIM2a, PLIM2b, and PLIM2c [89,92]. The roles of PLIM2a, PLIM2b, and PLIM2c have been investigated through RNA interference. The complete suppression of the three PLIM2s totally disrupted pollen development, producing abortive pollen grains and rendering the transgenic plants sterile. Their partial suppression arrested pollen tube growth to a lesser extent, resulting in short and swollen pollen tubes [93]. The knockout of PLIM2a resulted in short and broadened pollen tubes with defective actin bundles in the shank region [94]. These actin bundle defects could be rescued by PLIM2a as well as PLIM2b, suggesting a partially redundant function between PLIM2a and PLIM2b in organizing actin bundles in the shank [94]. PLIM2b co-localized with the long actin bundles along the pollen tubes, but was absent in the tip regions [94]. Differently, PLIM2c interacted with the long actin bundles of the shank region and occasionally decorated a subapical actin fringe-like structure [87]. Collectively, PLIM2s are important factors for *Arabidopsis* pollen development and tube growth [92,93].

### 3.5. CROLINs

The plant-specific CROLIN family contains one or two predicted actin cross-linking domains. There are six CROLIN members in *Arabidopsis* with a homology as high as 71%. CROLINl and CROLIN2 are specifically expressed in pollen, while CROLIN3-CROLIN6 are expressed in the vegetative organs. Biochemical analyses showed that CROLIN1 is a novel actin cross-linking protein with binding and stabilizing activities [95]. Moreover, CROLIN1 can cross-link actin bundles into actin networks [95]. In the crolin1 mutant, both pollen germination and pollen tube growth are significantly more likely to show perturbation by Latrunculin B (LatB) treatment. For instance, after treatment with LatB, the germination rate decreased, the growth rate slowed down, and the pollen malformation rate increased. These results proved that CROLIN1 engages in pollen germination and polarity growth of pollen tubes by regulating actin filaments [95].

### 3.6. ADFs

The ACTIN-DEPOLYMERIZING FACTOR (ADF/cofilin) family is an important class of low-molecular-weight actin-binding proteins that exists in all eukaryotes [96]. ADF was originally purified from embryonic chicken brain [97] and was subsequently shown to emerge as a central regulator of actin turnover in eukaryotes including plants [98,99,100]. The biochemical activities of ADF/cofilins have been documented extensively over the past several decades. ADF/cofilin can bind both ADP- and ATP-loaded G-actin, but it prefers ADP-loaded G-actin and inhibits nucleotide exchange [101,102]. Therefore, ADF/cofilin was thought to target older actin filaments [103]. Through its actin filament (AF) severing and pointed end-depolymerizing activities, ADF/cofilin enhances actin cytoskeleton dynamics [101,104,105].

In lily and tobacco pollen tubes, both green fluorescent protein (GFP)-ADF and immunocytochemistry with anti-ADF sera decorate actin filaments and show an accumulation of ADF in the cortical cytoplasm of the subapical region [17,106,107]. In tobacco, the overexpression of NtADF1 resulted in the reduction of fine, axially-oriented actin cables in the transformed pollen tubes and in the inhibition of pollen tube growth in a dose-dependent manner [17]. Thus, the proper regulation of actin turnover by NtADF1 is critical for pollen tube growth [17]. When expressed at a moderate level, green fluorescent protein (GFP)–tagged NtADF1 (GFP-NtADF1) associated predominantly with the subapical actin mesh and with long actin cables in the shank [17]. In lily pollen tubes, ADF specifically localizes at the actin fringe, and the modulation of intracellular pH profoundly alters the actin fringe as well as the distribution of ADF [107]. Cheung et al. used GFP-NtADF1, GFP-LlADF1, and NtPLIM2b-GFP as new actin reporters to re-confirm that the predominant actin structures in elongating tobacco and lily pollen tubes are actin cables along the pollen tube shank, and a subapical structure comprising shorter actin cables [108].

There are 11 ADF genes encoded by the *Arabidopsis* genome, and these genes can be divided into four subclasses [109]. Subclass I ADFs (AtADF1, AtADF2, AtADF3, AtADF4) are expressed at a relatively high level in all plant tissues except pollen [108]. Subclass II ADFs (AtADF7, AtADF8, AtADF10, AtADF11) are expressed specifically in mature pollen and pollen tubes or root epidermal trichoblast cells and root hairs, and are considered to be involved in a type of highly polarized growth activity that is regulated by the actin cytoskeleton [109,110]. Subclass III ADFs (AtADF5, AtADF9) are expressed weakly in vegetative tissues, but were the strongest in fast growing and/or differentiating cells [109]. Furthermore, subclass III evolved a F-actin-bundling function from a conserved F-actin-depolymerizing function [96]. The single subclass IV ADF (AtADF6) was constitutively expressed at moderate levels in all tissues, including pollen [109]. In subclass II, AtADF7 and AtADF10 are expressed specifically in pollen and pollen tubes [80,109]. Green fluorescent protein (GFP) fusions of both ADF7 and ADF10 were shown to decorate actin filaments and exhibit distinct localization patterns within pollen tubes [110]. Biochemical analyses revealed that ADF7 is a typical ADF that prefers ADP-G-actin over ATP-G-actin [103]. ADF7 inhibits nucleotide exchange on actin and severs filaments, but its filament severing and depolymerizing activities are less potent than those of the vegetative ADF1 [103]. ADF7 primarily decorates longitudinal actin cables in the shanks of pollen tubes [103]. In adf7 pollen tube shanks, the severing frequency and depolymerization rate of filaments significantly decreased, while their maximum lifetime significantly increased [103]. These results suggest that ADF-mediated severing regulates the turnover of longitudinal actin cables to promote pollen tube tip growth [103]. On the other hand, ADF10 was associated with filamentous actin in the developing gametophyte, in particular with the arrays surrounding the apertures of the mature pollen grain [110]. In the shank of elongating pollen tubes, ADF10 was associated with thick actin cables [110]. As atypical ADFs, both ADF5 and ADF9 exhibit a surprising ability to bundle and stabilize actin filaments in vitro [111,112]. The ADF5 and ADF9 expression patterns in mature pollen cells overlapped. A recent report showed that *Arabidopsis* ADF5 and ADF9 evolved a F-actin-bundling (B-type) function from a conserved F-actin-depolymerizing (D-type) function [112]. ADF5 is abundantly expressed in mature pollen, and plays an important role in pollen germination and pollen tube growth by forming and stabilizing higher-order actin structures [96]. There were some obvious defects in actin architectures in *adf5* pollen, which thus delayed the establishment of polarity [96]. The actin cytoskeleton in *adf5* pollen tubes was hypersensitive to LatB, and the skewness value was significantly decreased in *adf5* pollen grains, indicating that the extent of actin bundling was decreased due to ADF5 deficiency [96]. Accordingly, the loss of the ADF5 function led to a remarkable reduction in the length of the pollen tubes and the cytoplasmic streaming velocity in the tube cells became slower [96]. However, there was no obvious difference in pollen germination, pollen tube growth, and LatB sensitivity between the *adf5adf9* double mutant and the *adf5* mutant [96]. These data suggested that ADF5, but not ADF9, plays an important role in the maintenance and regulation of actin bundles during pollen germination and pollen tube growth [96].

## 4. Conclusions

So far, six different actin-binding proteins families have been studied using the model plant Arabidopsis as a research object. All of them have at least two homologous members in the pollen tube.

Are the homologous members performing the same function? In Figure 1, their role in the actin bundle construction is depicted. Some variants have redundant functions, such as AtVLN2 and AtVLN5, while others have a distinct cellular distribution, such as AtFH1 and AtFH5. Notably, the biochemical activities of some homologous members diverge, such as AtFIM4 and AtFIM5. AtFIM4 generates both actin bundles and branched actin filaments, whereas AtFIM5 generates only actin bundles [113]. Recently, our group has discovered AtFIM5 participating in the formation of actin bundles in the shank and also the tip, but AtFIM4 only functioned in the shank. Therefore, we propose that actin bundles in the shank may have different properties from those in the tip.

In the next decade, it is necessary to explore the properties of actin bundles of pollen tubes in detail, and to uncover how the six actin-binding proteins families corporately function to ensure proper actin bundle construction in pollen tubes.

## Figures and Tables

**Figure 1 ijms-19-03710-f001:**
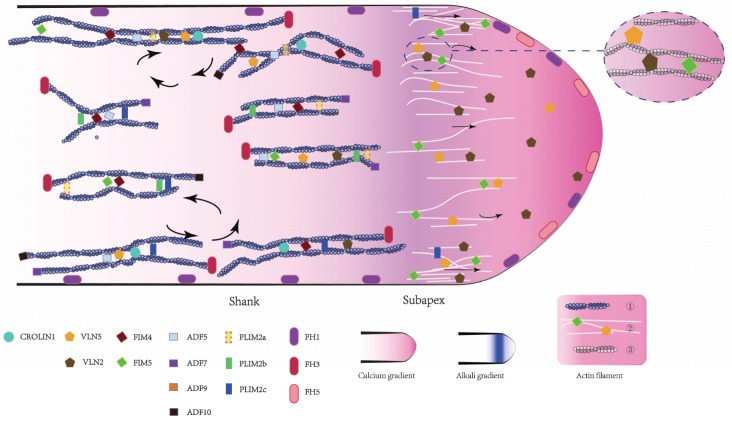
Schematic summary of the intracellular localization and function of actin-binding proteins in the *Arabidopsis* pollen tube. For detailed information regarding the functional characterization of each actin-binding proteins (ABP), see the description in the text.

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
