# Peer review of "Actin Bundles in The Pollen Tube"

_ijms, 2018, doi:10.3390/ijms19123710_

Round 1

Reviewer 1 Report

This is very usefull overview of the actin cytoskeleton in pollen tubes with focus on F-actin bundles.

The F-actin fringe (discussed in referencers 19 and 20) and shown in the Figure 1 is missing if the pollen-specific Actin3 is used for F-actin reporter expression (Jasik et al. 2016). Expression of the Lifeact reporter under control of the pollen-specific Actin3 promoter revealed 2 new aspects: (i) long F-actin bundles in pollen tube shanks are dynamic, showing undulating movements, (ii) subapical 'actin collars' or 'fringes' are absent

Jásik J et al. Actin3 promoter reveals undulating F-actin bundles at shanks and dynamic F-actin meshworks at tips of tip-growing pollen tubes. Plant Signal Behav 2016;11: e1146845

Authors should add this relevant information to their manuscript.

Author Response

We have added the relevant information to the manuscript and quoted the article. (see p2, line 73-78). Thank you very much for the valuable suggestions!

Reviewer 2 Report

This manuscript reviews recent advances in understanding a bundling mechanism of actin filaments in the pollen tube which is important for pollen growth and intracellular transport.

I have just two suggestions.

1. In this review, author describes functional redundancy and diversity of homologous members in bundling families. However, descriptions about their biological significance or role on pollen tube function is not enough. Author should mention about their contributions on pollen tube growth, guidance or transportation.

2. Schematic describing in Figure 1 is too simple to explain the relation between actin bundlers and actin in pollen tube described in this review. Author should revise them for better understanding by readers.

Author Response

1. We have added more information about the role of actin-bundling proteins on the pollen tube. (see p3, line 142-143; p4, line 174-177; 178-180; p5, line 244-245; p7, line 311-313).

2. Figure 1 has been modified as suggested (see p8, Fig.1).

Thank you very much for the valuable suggestions!